# Effects of 30-Day High-Dose Omega-3 Fatty Acid Supplementation on Plasma Oxidative Stress Enzyme Activities in Recreational and Trained Runners: A Pilot Study

**DOI:** 10.3390/nu17182985

**Published:** 2025-09-17

**Authors:** Bojan Martinšek, Milan Skitek, Tina Kosjek, Leon Bedrač, Evgen Benedik

**Affiliations:** 1Department of Food Science and Technology, Biotechnical Faculty, University of Ljubljana, 1000 Ljubljana, Slovenia; bojan.martinsek@gmail.com; 2University Institute of Clinical Chemistry and Biochemistry, University Medical Centre Ljubljana, 1000 Ljubljana, Slovenia; milan.skitek@kclj.si; 3Department of Environmental Sciences, Jožef Stefan Institute, 1000 Ljubljana, Slovenia; tina.kosjek@ijs.si; 4The NU B.V., J.H. Oortweg 21, 2333CH Leiden, The Netherlands; leon.bedrac@thenu.com; 5Division of Pediatrics, University Medical Centre Ljubljana, 1000 Ljubljana, Slovenia

**Keywords:** physical activity, running, reactive oxygen species, omega-3, polyunsaturated fatty acids, catalase, glutathione peroxidase, superoxide dismutase

## Abstract

**Background:** Physical activity induces the production of reactive oxygen species (ROS), which, at moderate levels mediate beneficial physiological adaptations, including insulin sensitivity and enhanced antioxidant defense. However, excessive ROS production during intense exercise may exceed endogenous antioxidant capacity, leading to oxidative stress and muscle damage. **Objective**: This study examined the effects of 30-day high-dose omega-3 fatty acid supplementation (9 g/day) on plasma fatty acid composition and the activity of antioxidant enzymes in recreational (*n* = 11) and trained (*n* = 10) runners, with emphasis on group- and time-specific responses. **Methods:** Plasma levels of arachidonic acid (AA), eicosapentaenoic acid (EPA), and docosahexaenoic acid (DHA) were assessed at three time points: pre-, during, and post-supplementation period. Enzymatic activities of glutathione peroxidase (GPx), superoxide dismutase (SOD), and catalase (CAT) were measured at six time points, including before and after exercise sessions involving a 2800 m run followed by a 400 m sprint. **Results:** Omega-3 supplementation increased plasma EPA and DHA. In trained runners, it was associated with a transient reduction in GPx and a pronounced mid-phase decline in SOD, whereas enzyme activities remained stable in recreational runners. CAT activity did not change significantly in either group. **Conclusions:** Short-term high-dose omega-3 supplementation modulates antioxidant enzyme activity in a group- and time-dependent manner. The observed downregulation of GPx and SOD in trained runners may reflect altered redox signaling; however, its relevance for exercise performance remains uncertain. Further studies are warranted to elucidate the physiological and functional consequences of these findings.

## 1. Introduction

The preponderance of scientific evidence consistently shows a favorable impact of moderate-intensity exercise training on both health span and lifespan [1]. These benefits are largely mediated by exercise-induced production of reactive oxygen species (ROS), which at physiological levels, trigger adaptive processes such as enhanced insulin sensitivity, mitochondrial biogenesis and upregulation of antioxidant defense systems [2,3,4,5]. In contrast, upregulated ROS production during strenuous exercise is associated with increased lipid peroxidation and protein carbonylation, elevated plasma creatine kinase activity and altered glutathione redox status, all of which can contribute to muscle damage [6,7,8]. High-intensity interval training, while recognized for its strong effects on metabolic rate and cardiorespiratory fitness [9], has been shown to markedly increase ROS and reactive nitrogen species, potentially surpassing the neutralizing capacity of endogenous antioxidant systems [3,10].

Oxidative stress delineates the state of an imbalance between oxidants and antioxidants in which oxidants predominate, potentially leading to damage of cellular DNA, proteins, lipids, and carbohydrates [11,12,13]. The concept of oxidative stress has often been redefined as a failure to control redox signaling [14], meaning that there are optimal levels of the responsible ROS and/or reactive nitrogen species required to maintain adaptive responses and cellular homeostasis [15,16]. If more pro-oxidants are produced than the antioxidant defense can handle, the balance of the antioxidant defense is lost in favor of free radical formation [17].

Numerous studies suggest that dietary supplementation with omega-3 fatty acids, characterized by the presence of the first double bond at the third carbon atom from the methyl end of the fatty acid molecule, can stimulate the activation of endogenous enzymatic antioxidant systems that neutralize ROS. These include enzymes that eliminate peroxides, such as the intracellular enzymes glutathione peroxidase (GPx), peroxiredoxins and catalase (CAT), as well as those that neutralize superoxide, such as superoxide dismutase (SOD), which forms the first line of defense in erythrocytes [14,18]. SOD catalyzes the conversion of superoxide radicals to hydrogen peroxide (H_2_O_2_), which is subsequently degraded to water by CAT and GPx [18]. GPx plays a central role in eliminating peroxides, maintaining the thiol/disulphide redox state of proteins, and ensuring the reduced and functional forms of ascorbate and (indirectly) vitamin E, thereby serving as a key antioxidant [14].

Importantly, the same structural features of omega-3 fatty acids that confer biological activity—multiple double bonds—also render them highly susceptible to lipid peroxidation. This duality raises concerns that high intakes may paradoxically increase oxidative stress [19]. Thus, omega-3 fatty acids act as a double-edged sword, with both antioxidant potential and pro-oxidative risk depending on context and dosage.

A variety of omega-3 fatty acids occur in nature. However, most scientific research has been centered around eicosapentaenoic acid (EPA), docosahexaenoic acid (DHA) and alpha-linolenic acid (ALA), as they seem to be the most relevant to human physiology and metabolism [20]. ALA is present in both plants and terrestrial animals, whereas EPA and DHA derive primarily from seafood and algae. Although humans can synthesize EPA and DHA de novo via enzymes involved in the elongation and desaturation of a plant-derived, shorter-chain omega-3, ALA, this pathway is notably inefficient. In fact, the conversion in most healthy adults is only about 5 to 10 percent to EPA and 2 to 5 percent to DHA [21,22]. Many observational studies have consistently shown that marine sources of omega-3 have many beneficial health effects, particularly cardio- and neuroprotective roles, while also reducing the risk of developing diabetes and metabolic syndrome [20,23,24]. Nevertheless, there is also some controversy about their efficacy and certain human health benefits, apparently due to the different dosages used in studies [25].

The beneficial effects of fish oil-derived omega-3 on the body’s antioxidant defense system appear to be mediated, at least in part, through activation of nuclear factor erythroid 2 related factor 2 (Nrf2) [26,27,28]. However, the influence of highly unsaturated fatty acids on redox homeostasis remains debated, as high consumption of omega-3 fatty acids leads to their accumulation in cell membranes and tissues, rendering these cellular components more susceptible to lipid peroxidation due to the presence of highly reactive conjugated double bonds [19]. One of the most prominent biochemical functions of omega-3 fatty acids is their anti-inflammatory effect. The latter is mediated via the pro-inflammatory prostaglandin E2 induced by omega-6 fatty acids, the reduction in the pro-inflammatory master transcriptional regulator nuclear factor-κB, and the increase in anti-inflammatory resolvin and protectin levels, rather than via direct inhibition of ROS [29]. Peroxidation of omega-3 polyunsaturated fatty acids (PUFAs) within mitochondria may transiently increase ROS, yet the same process also indirectly reduces oxidative stress by upregulation of antioxidant defense enzymes such as SOD [29].

Adequate intake of omega-3 PUFAs has been reported to attenuate of cytokine and ROS production [30,31]. However, supplementation per se does not mitigate exercise-induced oxidative stress, and may even exacerbate resting oxidative stress [32]. Consequently, the cumulative effect of omega-3 PUFAs on oxidative stress during increased physical activity remains equivocal and remains the subject of ongoing debate. Heterogeneity in exercise protocols, supplementation strategies (dosage, duration, timing), and population characteristics (training status, fitness level) likely contribute to the inconsistencies in the outcomes across investigations [33].

Recently, a randomized controlled trial examined the effect of a 3-week low-dose omega-3 supplementation on inflammatory adipocytokines and blood antioxidant defense capacity in non-elite endurance runners [34]. A pilot study was also conducted to investigate the effect of a moderate dose (i.e., 4 g/day) of omega-3 supplementation on physical exercise and markers of oxidative stress and inflammation in middle- and long-distance runners, comparing athletes with sedentary subjects [35]. The authors concluded that including a third group of recreational athletes would provide additional insight into the still elusive physiological role of high doses of omega-3 on antioxidant defense systems in vivo.

To the best of our knowledge, no previous study has specifically investigated the effect of high-dose fish oil supplementation on oxidative stress enzymes in athletes. However, related research has examined lower doses of omega-3 in athletes [34,35] and high-dose regimens in certain clinical populations (e.g., inflammatory and metabolic disorders), which have provided mixed results. Acknowledging this broader body of evidence highlights the novelty of the present study in exploring a high-dose intervention in trained and recreational runners.

## 2. Materials and Methods

### 2.1. Participants

Twenty-four out of forty volunteers agreed to participate and met the inclusion criteria, which were as follows: (1) healthy adult males aged 18–39 years (women were excluded to avoid confounding effects of sex hormones on oxidative stress markers and because of the small sample size); (2) a body mass index (BMI) between 20 kg/m^2^ and 30 kg/m^2^; (3) no intake of omega-3 supplements; (4) no history of physical or mental illness; (5) classification into one of two categories: (a) recreational runners who engaged in running at least 45 min three times a week; or (b) trained runners registered with the Slovenian Athletics Federation and participating in national or international competitions.

The sample size was determined primarily by feasibility considerations, including the availability of well-characterized trained runners and the demanding nature of the supplementation and testing protocol.

All participants were fully informed about the experimental protocols and signed an informed consent form, authorizing the experimenter to use the results for scientific publication purposes. The study protocol was approved by the National Medical Ethics Committee of Republic of Slovenia (identification number KME 85/09/14; date of approval: 16 September 2014).

### 2.2. Study Design

The study design consisted of three phases, as shown in Figure 1, and was conducted in Ljubljana, Slovenia, between September and November 2013: (1) a pre-supplementation period encompassing baseline measurements prior to, 24 h after, and 48 h after the initial running-induced stress test (Run1); (2) a 30-day supplementation period involving the intake of 9 g omega-3 per day; and (3) a post-supplementation period with measurements prior to, 24 h after, and 48 h after the second running-induced stress test (Run2). In both cases of the running-induced stress tests (Run1 and Run2), the protocol included a 2800 m run followed by an all-out sprint during the final 400 m. Anthropometric measurements (height and weight) were recorded to calculate BMI; all blood samples were collected at the University Medical Centre Ljubljana, Slovenia. Fasting venous blood was collected by antecubital venipuncture using Vacutainer tubes (Becton Dickinson, Franklin Lakes, NJ, USA). Lavender-cap K_2_EDTA tubes were used for fatty acid profiling (AA, EPA, DHA) on Day 1 (T0), Day 24 (mid-supplementation; the 19th day of the 30-day supplementation phase), and Day 36 (T0′). Green-cap lithium heparin tubes were used for enzyme assays (SOD, GPx, CAT) on Day 1 (T0), Day 2 (T24), Day 3 (T48), Day 36 (T0′), Day 37 (T24′), and Day 38 (T48′). Tubes were gently inverted according to the manufacturer’s instructions. Plasma was prepared following standard laboratory procedures, and aliquots were stored at −80 °C until analysis at the Jožef Stefan Institute (Ljubljana, Slovenia).

These time points were selected to capture the delayed and recovery-phase responses of antioxidant enzyme activities, which are considered relevant for understanding training adaptations. Although this approach provided valuable information on longer-term redox regulation, it may have missed acute oxidative stress responses occurring within the first few hours post-exercise.

Three days following the running-induced stress test, all participants initiated a dietary intervention involving daily supplementation with 9 g of omega-3. Each participant received 360 gelatin capsules containing the omega-3 (Omega-3 RX Food Supplement; Masel Ltd., Ljubljana, Slovenia), whose fatty acid composition is regularly assessed by the International Program of Fish Oil Standards administered by Nutrasource Diagnostics, Inc. (Guelph, ON, Canada). Each capsule contained 750 mg of omega-3, consisting of 400 mg EPA, 200 mg DHA, 150 mg of other omega-3 fatty acids and 2.8 mg of tocopherols. Each participant took 12 capsules of the supplement as a single morning bolus. This bolus intake was selected to maximize compliance and reduce the risk of missed doses. Similar bolus protocols have been used in supplementation studies to ensure adherence [36]. This provided a total daily intake of 4.80 g EPA, 2.40 g DHA, 1.80 g of other omega-3 fatty acids, and 33.6 mg of mixed tocopherols. Throughout the 30-day supplementation period, all participants were contacted every other day to reinforce compliance with supplementation and monitor any potential side effects. On day 19 of supplementation period, the fourth blood sample was taken from all participants and analyzed for plasma AA, EPA and DHA levels. The latter served only as a control measurement to ensure adherence to the supplementation protocol. Subsequently, a day after the supplementation period concluded (Day 36; Figure 1), the same protocol of measurements and running-induced stress test used during the pre-supplementation period (starting with day 1) was repeated. This approach yielded paired data at T0′, T24′ and T48′ in comparison to T0, T24 and T48, respectively.

Adopting an ecologically valid protocol that combines a prolonged endurance run (2800 m) with a maximal sprint (400 m) reflects the combined physiological stresses experienced by runners in real competition or training and aligns with the push in the recent literature to diversify exercise models to better interrogate oxidative stress dynamics.

### 2.3. Gas Chromatography–Mass Spectrometry Analysis for Plasma AA, EPA, and DHA Levels

The gas chromatograph employed (HP 6890 series; Hewlett Packard, Waldbronn, Germany) was interfaced with a single quadrupole mass-selective detector (HP 6890 series; Hewlett Packard, Waldbronn, Germany). A capillary column (DB-5 MS; 30 m × 0.25 mm × 0.25 mm; Agilent J&W, Santa Clara, CA, USA) using helium as the carrier gas (37 cm·s^−1^), was utilized. One-microliter samples were injected into the system at 250 °C in splitless mode, and the transfer line was maintained at 280 °C. The mass spectrometer operated in electrospray ionization mode at 70 eV. Employing the selected ion monitoring mode, the following ions were monitored (quantification ions in bold): AA, *m*/*z* 203, 220, 247; EPA, *m*/*z* 201, 220, 247; DHA, *m*/*z* 159, 215, 241; and AA-d8 (Sigma Aldrich, St. Louis, MO, USA) internal standard, *m*/*z* 209, 225, 255. Chemstation software (Version E.02.00, Agilent, Santa Clara, CA, USA) was used for instrument control and data processing for the gas chromatography mass-selective detector. Prior to analysis, 50 µL plasma samples were subject to derivatization at 90 °C for 10 min with 3 mL 0.5 M NaOH and 3 mL boron trifluoride/20% methanol mixture (Merck Kenilworth, Rahway, NJ, USA).

### 2.4. Enzyme Activity Assays

To determine GPx activity, glutathione was oxidized by cumene hydroperoxide in the presence of GPx. In the second step, oxidized glutathione was converted to its reduced form by glutathione reductase and NADPH, simultaneously oxidizing NADPH to NADP+. The reduction in absorbance resulting from the consumption of NADPH was measured at 340 nm (Olympus AU400 analyzer; Mishima Olympus Co., Ltd., Tokyo, Japan; reagent: Randox Laboratories Limited, Crumlin, UK).

The SOD activity was measured by the degree of inhibition of the formazan dye generated by superoxide radicals formed from xanthine in the presence of xanthine oxidase. The detection of the resulting product was performed at 505 nm (Olympus AU400 analyzer; Mishima Olympus Co., Ltd., Tokyo, Japan; reagent: Randox Laboratories Limited, Crumlin, UK).

CAT activity was determined in a two-step procedure. In the first step, H_2_O_2_ underwent dismutation to water in the presence of CAT, and in the second step, the residual H_2_O_2_ was quantified by formation of the quinoneimine dye in the presence of horseradish peroxidase, with detection at 520 nm (Olympus AU400 analyzer; Mishima Olympus Co., Ltd., Tokyo, Japan; reagent: OxisResearch, Foster City, CA, USA).

### 2.5. Statistical Analysis

The analysis was performed using R (version 4.4.1; R Foundation for Statistical Computing, Vienna, Austria). Data import and initial cleaning were handled with the readxl and janitor packages. The tidyverse suite of packages was employed for general data manipulation and visualization.

Descriptive statistics, including mean, standard deviation, median, and range, were calculated for baseline characteristics and key outcome variables (AA, EPA, DHA) for the entire cohort and stratified by group (recreational vs. trained runners) and time point.

To analyze the changes in oxidative stress biomarkers (GPx, SOD, and CAT), robust linear mixed-effects models were used, fitted with the robustlmm package. This approach was chosen to minimize the influence of potential outliers in the data. The models included fixed effects for group (recreational, elite), time (T0, T24, T48, T0′, T24′, T48′), and the group-by-time interaction. A random intercept for each participant (id) was included to account for the non-independence of repeated measures.

Following the model fitting, post hoc analyses were conducted using the emmeans package to calculate estimated marginal means. This allowed for pairwise comparisons both between groups at each time point and within groups across different time points. *p*-Values for these multiple comparisons were adjusted using the Holm method to control the family-wise error rate.

## 3. Results

During the study, one recreational runner withdrew consent and was excluded. Additionally, two participants—one from each group—were removed from the final analysis due to non-compliance with the supplementation protocol, as reflected by unchanged plasma EPA and DHA levels between Day 1 and Day 19, indicating that supplementation had not been taken as instructed. The final dataset therefore comprised 21 participants: 11 recreational and 10 trained runners. Baseline anthropometric characteristics are shown in Table 1. No significant differences were observed between recreational and trained runners; however, trained runners tended to be younger (22.7 vs. 27.9 years), lighter (70.1 vs. 72.2 kg), and had a lower BMI (21.2 vs. 22.7 kg/m^2^). These parameters remained stable throughout the omega-3 supplementation period.

As shown in Table 2, plasma AA did not change significantly in any group after 30 days of omega-3 supplementation. EPA increased overall (+53.5%; 159.1 vs. 244.1 µg/mL; *p* = 0.004); within sub-groups, EPA rose in recreational runners (+69.0%; 152.9 vs. 258.3 µg/mL; *p* = 0.032) and showed a non-significant increase in trained runners (+37.8%; 165.8 vs. 228.5 µg/mL; *p* = 0.073). DHA increased overall (+174.6%; 32.2 vs. 88.5 µg/mL; *p* < 0.001); by subgroup, the increase was significant in recreational runners (+289.2%; 24.3 vs. 94.7 µg/mL; *p* = 0.030) and not significant in trained runners (+99.6%; 40.9 vs. 81.7 µg/mL; *p* = 0.080).

The temporal profiles of GPx, CAT, and SOD are shown in Figure 2, Figure 3 and Figure 4.

For GPx, there were no significant differences between groups at individual time points (all *p* ≥ 0.20). Mean activity at baseline (T0) was 0.96 μkat·g^−1^ (95% CI: 0.81–1.11) in recreational group, and 1.10 μkat·g^−1^ (0.94–1.26) in trained group. In the recreational group, GPx activity remained stable across all time points (all Holm-adjusted *p* = 1.00). In the trained group, however, a transient decline was observed at T0′ (0.90 μkat·g^−1^, 95% CI: 0.74–1.06), which was significantly lower than value at T24 (1.17 μkat·g^−1^, Δ = 0.268; *p* = 0.007), T24′ (1.15 μkat·g^−1^, Δ = 0.246; *p* = 0.019), and T48′ (1.20 μkat·g^−1^, Δ = 0.302; *p* < 0.001).

CAT activity was generally comparable between groups and consistent over time. At T0, values were 2.94 μkat·g^−1^ (2.53–3.36) in recreational group and 2.86 μkat·g^−1^ (2.41–3.31) in trained group. The only significant difference occurred at T24, when trained group displayed higher CAT activity (3.07 μkat·g^−1^, 2.62–3.52) compared with recreational group (2.30 μkat·g^−1^, 1.87–2.73), with a mean difference of 0.77 μkat·g^−1^ (*p* = 0.015). No other between-group differences reached significance (all *p* ≥ 0.26), and within-group contrasts did not reveal meaningful changes over time.

In the case of SOD, more pronounced differences emerged. At baseline, trained group exhibited significantly higher activity (24.7 μkat·g^−1^, 23.2–26.3) compared with recreational group (22.4 μkat·g^−1^, 21.0–23.9), with a mean difference of 2.29 μkat·g^−1^ (*p* = 0.034). In recreational group, SOD activity remained relatively stable throughout the protocol, with values ranging from 19.9 to 22.7 μkat·g^−1^ (all adjusted *p* ≥ 0.26). In contrast, the trained group showed a marked decline after T0. At T0′, mean activity decreased to 19.7 μkat·g^−1^ (18.2–21.3), which was significantly lower than baseline (Δ = 4.98; *p* < 0.001). At T24′, activity decreased further to 18.9 μkat·g^−1^ (17.4–20.5), representing a difference of 5.81 μkat·g^−1^ compared with T0 (*p* < 0.001) and reductions relative to both T24 (22.9 μkat·g^−1^; Δ = 3.98; *p* = 0.005) and T48 (23.4 μkat·g^−1^; Δ = 4.46; *p* < 0.001). By T48′, SOD activity had partially recovered (20.6 μkat·g^−1^, 19.1–22.2) but remained lower than baseline (Δ = 4.09; *p* = 0.003).

In summary, GPx activity was stable in recreational group but transiently suppressed at T0′ in trained group, CAT was unchanged except for higher activity in trained group at T24, and SOD was initially elevated in trained group but declined sharply during the mid-phase, with the lowest values at T24′, while remaining stable in recreational group.

## 4. Discussion

The present study investigated the effect of 30-day high-dose omega-3 supplementation on oxidative enzyme activities in recreational and trained runners, both at rest and during recovery (24 h and 48 h) following a running-induced stress test.

To our knowledge, this is the first study to examine the influence of high-dose omega-3 fatty acids supplementation on antioxidant enzyme responses in these two populations under conditions of acute exercise stress. Thirty days of omega-3 PUFA supplementation modified plasma activities of key antioxidant defense enzymes, with more pronounced effects observed in trained compared to recreational runners. GPx activity remained stable in recreational runners but showed a transient reduction at T0′ in trained runners, while SOD activity declined markedly during the mid-phase in trained runners but stayed unchanged in recreational runners. CAT activity was largely unaffected, aside from a single increase in trained runners at T24.

These results suggest potential group-specific variability in antioxidant enzyme regulation, possibly reflecting differences in metabolic adaptations associated with training status. While the reduction in enzyme activity is clear, its interpretation in terms of training adaptations remains speculative. The existing literature highlights a complex interplay between antioxidant modulation, exercise-induced ROS, and adaptive signaling. Some degree of oxidative stress is necessary to promote beneficial physiological adaptations, whereas excessive suppression of ROS or antioxidant activity may blunt these processes. Further studies that directly assess performance outcomes and molecular markers of adaptation are needed to clarify whether the observed downregulation in enzyme activity contributes positively or negatively to training responses.

Oral omega-3 supplementation is known to elevate plasma and membrane levels of EPA and DHA, often accompanied by a decrease in omega-6 arachidonic acid due to their substitution. Notably, a study by Arnold et al. [36] investigated a 12-week supplementation of 1.6 g/day of EPA and DHA in a ratio of 7:1. They reported a sevenfold increase in plasma EPA levels and a nearly 50% increase in DHA levels, along with a 12% reduction in AA levels. In our study, the post-supplementation plasma AA levels did not show significant decrease in either group of runners. Instead, AA levels increased by 26.4% in recreational runners and only by 2.5% in trained runners. These changes were accompanied by a significant increase in both EPA and DHA, with DHA levels nearly tripling in recreational runners and doubling in trained runners. While the EPA levels were notably higher in recreational runners, a significant increase was not observed in trained runners (though a positive trend was evident). The disparity in EPA increase between our study and that of Arnold et al. [36] is most likely attributable to the different EPA:DHA ratios applied (2:1 vs. 7:1).

Moreover, our study revealed altered antioxidant systems among runners. At baseline trained runners exhibited higher SOD activity (~8%; 24.7 vs. 22.9 μkat·g^−1^), comparable GPx activity (1.10 vs. 0.98 μkat·g^−1^; +12.3%), and lower CAT activity (2.85 vs. 3.04 μkat·g^−1^; −6.3%) relative to recreational runners. Mean baseline GPx values were comparable between groups, whereas SOD was ~8% higher in trained runners (24.7 vs. 22.9 μkat·g^−1^). These findings align with previous reports showing that trained athletes from various sports, such as competitive runners [37], cyclists [38], football [39], rugby [40], and handball [41], generally demonstrate enhanced resting antioxidant enzyme activities compared with untrained or sedentary individuals. Conversely, one study reported lower antioxidant enzyme activities at rest in trained compared with recreational cyclists [10]. Such discrepancies may be reconciled by the hormesis framework, which proposes a dose–response relationship whereby regular exposure to exercise-induced oxidative stress elicits adaptive upregulation of antioxidant defenses [42]. This concept extends to physical exercise [43,44] with intense physical exertion yielding greater ROS production in unconditioned individuals [3,45] and heightened antioxidant adaptation in highly conditioned athletes [46], thereby elucidating the observed differences in antioxidant enzyme responses.

Adequate intake of omega-3 has been associated with reduced cytokine production and ROS generation [30,31]. However, the supplementation does not appear to mitigate the exercise-induced oxidative stress, and it may even increase oxidative stress at rest due to lipid peroxidation [32]. The possibility that omega-3 supplementation could compromise training effects through altered SOD, GPx, and CAT activities is intriguing but remains speculative without performance outcomes. While changes in antioxidant enzyme activities provide valuable insight into redox response to supplementation, these biochemical findings should not be equated with functional improvements or impairments in exercise performance. As our study did not directly measure performance metrics, the link between reduced antioxidant enzyme activity and training adaptations should be interpreted with caution.

In a study by Groussard et al. [47], erythrocyte GPx and SOD activities were measured at multiple points at rest and up to 40 min post the Wingate test in eight healthy male volunteers. While GPx activity did not change significantly, a trend of increase was noted at the 40 min mark. In contrast, Fisher et al. [48] documented significant post-exercise increases in GPx activity within lymphocytes after high-intensity interval training. GPx activity returned to baseline levels three hours post-exercise and maintained this level for the ensuing 24 h.

In the present study GPx activity was not significantly altered overall; however, a transient suppression was observed at T0′ in trained runners, followed by recovery at later time points. Additionally, baseline GPx after supplementation decreased, aligning more closely with recreational levels. This may be attributed to the fact that GPx activity had already returned to pre-exercise levels. This finding suggests that high-dose omega-3 intake can reduce GPx activity to levels characteristic of less conditioned individuals, thereby potentially attenuating exercise-induced oxidative stress. Because such stress is considered essential for driving adaptive responses that enhance performance, excessive omega-3 supplementation might impair training-induced adaptations, an effect reminiscent of that reported for antioxidant supplementation [4,49,50]. Interestingly, this omega-3-induced decline in baseline GPx levels was not evident at 24 h or 48 h post-exercise, despite the expected increase in ROS production following the running challenge. When ROS generation exceeds the neutralizing capacity of the antioxidant defense systems, oxidative damage occurs in the muscle, liver, blood, and potentially in other tissues [31,37,51,52,53].

Among the antioxidant enzymes, SOD stands out as one that has been shown to exhibit alterations in response to both chronic and acute exercise [41,45,47,54]. Groussard et al. [47] demonstrated a notable decline in SOD activity at the onset of the Wingate test, followed by a gradual recovery that did not reach pre-test levels. The rapid decrease in SOD activity suggests the presence of H_2_O_2_, which inhibits SOD activity. Studies have indicated that baseline SOD activities are higher in well-conditioned cyclists and amateur athletes compared to controls. Notably, SOD activity increases significantly after prolonged cycling races, as reported by Mena et al. [55]. Contrary to this, Fisher et al. [48] reported that SOD activity was elevated immediately and up to three hours post-exercise, returning to pre-exercise levels after 24 h. Our analysis of antioxidant defense enzyme activities revealed that omega-3 fatty acid supplementation exerted the most pronounced effect on SOD activity in trained runners. This aligns with the role of SOD as an early defense against ROS in erythrocytes and mitochondria [51]. In trained runners, SOD activity was higher at baseline compared to recreational runners but declined significantly at T0′, T24′, and T48′, whereas recreational athletes maintained relatively stable SOD levels. This decrease in SOD activity may reflect a downregulation of SOD expression, potentially reducing the requirement for acute upregulation during oxidative stress. Alternatively, it could indicate suppression of antioxidant capacity or alterations in redox balance. This remains an open question that requires further investigation to clarify the role of omega-3 supplementation in modulating antioxidant defense systems.

Our interpretation contrasts with the viewpoint of Mandelsohn and Larrick, who proposed that omega-3 fatty acids enhance ROS regulation indirectly by upregulation oxidative defense enzymes, including SOD [29]. The discrepancy between these finding may reflect differences in study design, supplementation protocols, or training status of participants, underscoring the need for mechanistic studies to delineate how omega-3 fatty acids influence redox homeostasis in athletes.

In the present study, no consistent or significant changes in CAT activity were observed across supplementation or recovery, except for a transient increase in trained runners at T24 that did not persist. This may reflect insufficient oxidative stress to elicit discernible shifts in CAT activity. While GPx and CAT are both involved in H_2_O_2_ elimination, their substrate affinities differ: GPx is more effective at lower H_2_O_2_ concentrations, whereas CAT predominates under higher H_2_O_2_ levels. Another possible explanation is related to sampling design. In our study, the participants performed a 2800 m run that ended with a 400 m all-out sprint, with blood samples collected at 24 h and 48 h post-exercise. These time points may have missed the immediate oxidative stress response, potentially explaining the lack of detectable CAT changes. However, we also acknowledge that the study design may not have been sufficiently sensitive to capture more subtle or transient alterations in CAT activity. In the study by Michailidis et al. [56], oxidative stress was induced by an acute treadmill run at 90% of maximal oxygen uptake (VO2max), with samples collected multiple times within a 24 h period. They reported a substantial surge in CAT activity immediately post-exercise, which returned to baseline levels within two hours. In contrast, our study did not reveal significant differences. Similar observations in CAT activity were reported by Djordjevic et al. [41], where 58 young handball players and 37 controls completed a maximal progressive cycling test, resulting in decreased CAT activity immediately post-exercise in the athletes. Given these considerations, it is perhaps less surprising that our post-exercise sampling at 24 h and 48 h did not capture significant fluctuations in CAT activity.

While our findings highlight potential roles of omega-3 supplementation in modulating antioxidant enzyme activities, they should be interpreted with caution. The absence of a placebo-controlled group makes it difficult to attribute the observed changes solely to omega-3 supplementation. The relatively small sample size (*n* = 21), particularly once divided into recreational and trained subgroups, further reduces statistical power and increases the risk of type I and type II errors, meaning that some findings may reflect chance while others may have gone undetected. Similarly, the choice to collect blood samples only at 24 h and 48 h post-exercise focused our analysis on the recovery phase but may have overlooked acute oxidative stress responses occurring in the hours immediately after exertion. Finally, the non-standardized nature of the exercise protocol (2800 m run with a 400 m sprint) limits comparability with other studies. Taken together, these factors indicate that our results should be regarded as preliminary and hypothesis-generating. Replication in well-powered, placebo-controlled trials with standardized exercise protocols and direct assessment of performance outcomes is required.

## 5. Conclusions

In conclusion, 30 days of high-dose omega-3 supplementation reduced the activity of the antioxidant enzymes SOD and GPx but not CAT. This effect was more pronounced in trained runners, with both enzyme activities showing a decrease under non-stressed conditions. Our findings suggest that omega-3 supplementation may modulate redox signaling pathways involved in exercise-induced adaptations. However, these interpretations should be made with caution, as this pilot study had a small sample size and did not assess performance outcomes, limiting direct conclusions about the functional relevance of the observed biochemical changes.

Future research should include larger randomized, placebo-controlled trials to confirm these findings and elucidate the impact of high-dose omega-3 supplementation on oxidative stress and athletic performance. Moreover, a more comprehensive assessment of oxidative stress markers—such as malondialdehyde, 4-hydroxynonenal, and 8-hydroxydeoxyguanosine—as well as their associated mineral cofactors (e.g., Se, Fe, Zn, Cu, Mn) is warranted to provide deeper mechanistic insight into the interplay between supplementation, redox status, and training adaptations.

## 6. Limitations

This study has several limitations. First, the male-only design limits generalizability and is noted as a key limitation of the study. While we aimed to control for potential confounding factors, the exclusion of females may reduce the applicability of our findings to a broader population. Second, although volunteers were instructed not to consume other dietary supplements or exercise excessively during the experiment, complete compliance could not be guaranteed. Third, the study was not randomized and not placebo-controlled, which prevents definitive attribution of the observed effects solely to omega-3 supplementation. Fourth, the final sample size was small (*n* = 21) and dividing participants into recreational and trained subgroups further reduced statistical power and increased the risk of type I and type II errors. Fifth, blood sampling was performed only at 24 h and 48 h post-exercise, which may have missed the acute oxidative stress response. Sixth, the exercise test protocol (2800 m run with a 400 m sprint), while designed for ecological validity, is not standardized in the literature, which may reduce the reproducibility of our findings across different studies. Seventh, the bolus intake of omega-3 supplementation was selected to maximize compliance and reduce the risk of missed doses. While similar bolus protocols have been used in supplementation studies to ensure adherence [36], this design may influence absorption kinetics, which should be considered when interpreting the results. These limitations should be carefully considered when interpreting our results, and future research should aim to address them by conducting larger, adequately powered, randomized, placebo-controlled trials with standardized exercise protocols, more frequent blood sampling, and attention to gender diversity in study design.

## Figures and Tables

**Figure 1 nutrients-17-02985-f001:**
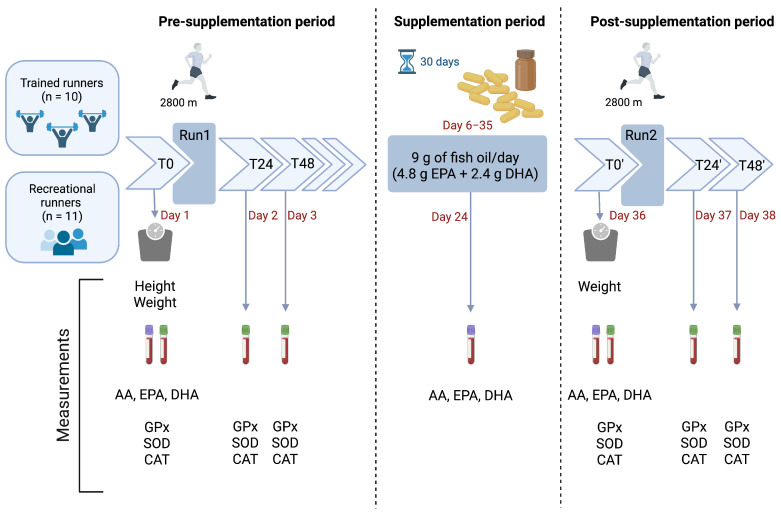
Design of the experimental protocol. Run1/Run2—Running-induced stress tests (2800 m run + 400 m sprint). T0, T24, T48—Time points before, 24 h after, and 48 h after Run1; T0′, T24′, T48′—Time points before, 24 h after, and 48 h after Run2. Fasting venous blood was collected by antecubital venipuncture using Vacutainer tubes (Becton Dickinson, Franklin Lakes, NJ, USA): a lavender-cap K_2_EDTA tube for fatty acid profiling (AA, EPA, DHA) collected on Day 1 (T0), Day 24 (mid-supplementation; the 19th day of the 30-day supplementation period), and Day 36 (T0′); a green-cap lithium heparin tube for enzyme assays (SOD, GPx, CAT) collected on Day 1 (T0), Day 2 (T24), Day 3 (T48), Day 36 (T0′), Day 37 (T24′), and Day 38 (T48′). AA—Arachidonic acid; CAT—Catalase; DHA—Docosahexaenoic acid; EPA—Eicosapentaenoic acid; GPx—Glutathione peroxidase; SOD—Superoxide dismutase. Created with BioRender.com.

**Figure 2 nutrients-17-02985-f002:**
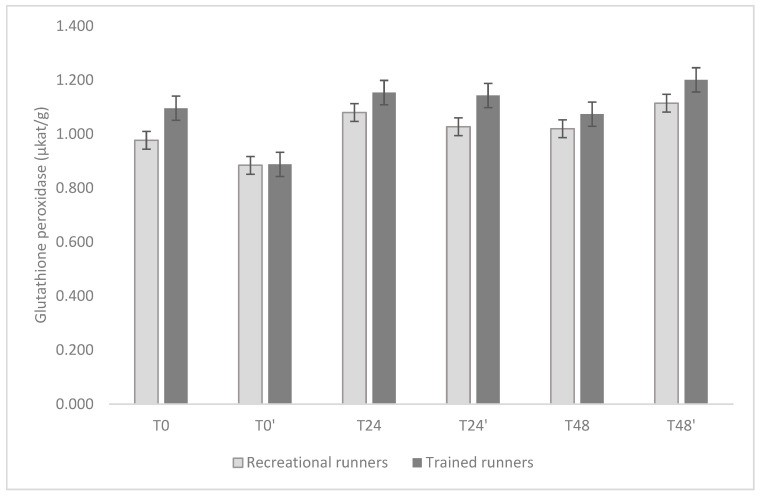
Comparisons of omega-3 supplementation effects (days 6–35 inclusive) on plasma glutathione peroxidase activity (days T0, T0′) and 24 h (days T24, T24′) and 48 h (days T48, T48′) after the running-induced stress test, for all participants. Data are presented as mean ± standard deviation.

**Figure 3 nutrients-17-02985-f003:**
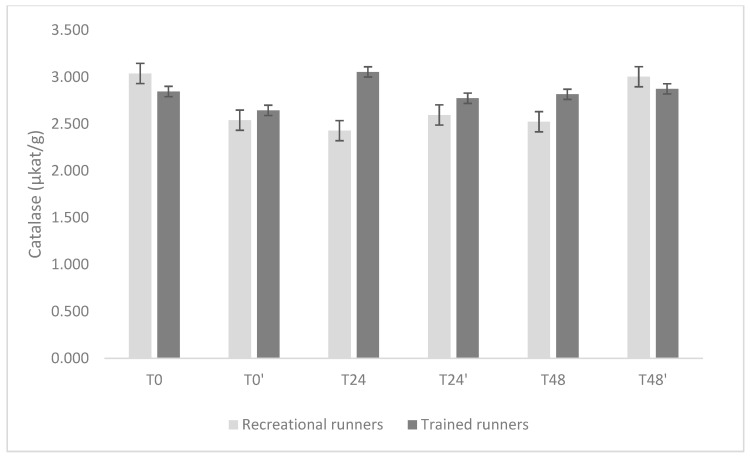
Comparisons of omega-3 supplementation effects (days 6–35 inclusive) on catalase activity before (days T0, T0′) and 24 h (days T24, T24′) and 48 h (days T48, T48′) after the running-induced stress test, for all participants. Data are presented as mean ± standard deviation.

**Figure 4 nutrients-17-02985-f004:**
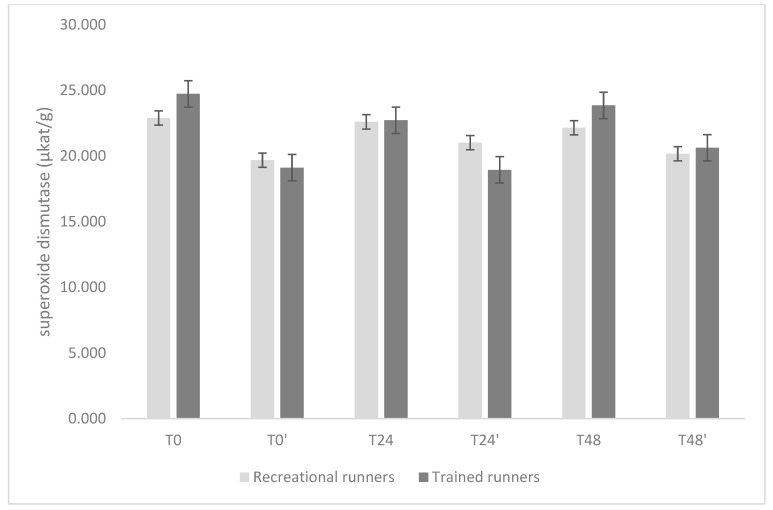
Comparisons of omega-3 supplementation effects (days 6–35 inclusive) on plasma superoxide dismutase activity before (days T0, T0′) and 24 h (days T24, T24′) and 48 h (days T48, T48′) after the running-induced stress test, for all participants. Data are presented as mean ± standard deviation.

**Table 1 nutrients-17-02985-t001:** Baseline anthropometric data recorded before (T0, Day 1) and after (T0′, Day 36) 30 days of omega-3 fatty acids supplementation, for all participants and recreational and trained runners separately.

Runner Group	Omega-3 Supplementation Period	Age (Years)	Height (cm)	Weight (kg)	Body Mass Index (kg/m^2^)
All (*n* = 21)	T0	25.29 ± 5.99	180.05 ± 5.91	71.03 ± 7.38	21.94 ± 2.26
	T0′	nd	nd	71.59 ± 7.90	22.10 ± 2.40
Recreational (*n* = 11)	T0	27.91 ± 4.70	178.55 ± 6.15	72.23 ± 7.67	22.65 ± 1.86
	T0′	nd	nd	72.86 ± 8.37	22.93 ± 2.10
Trained (*n* = 10)	T0	22.67 ± 5.73	181.92 ± 6.27	70.10 ± 6.71	21.23 ± 2.32
	T0′	nd	nd	70.27 ± 6.68	21.31 ± 2.33

Data are presented as mean ± standard deviation (SD). nd—not determined; T0—pre-supplementation period; T0′—post-supplementation period.

**Table 2 nutrients-17-02985-t002:** Plasma arachidonic acid (AA), eicosapentaenoic acid (EPA), and docosahexaenoic acid (DHA) between recreational and trained runners before (T0, Day 1) and after (T0′, Day 36) 30 days of omega-3 fatty acids supplementation.

Runner Group	Omega-3 Supplementation Period	Statistic	AA (µg/mL)	r	*p*	EPA (µg/mL)	r	** *p* **	**DHA** **(µg/mL)**	**r**	** *p* **
All participants (*n* = 21)	T0	Mean ± SD	158.60 ± 62.06	−0.26	0.31	159.05 ± 80.60	−0.68	0.001	32.22 ± 18.82	−0.68	< 0.001
		Median (Q1–Q3)	137.45 (119.63–175.95)			160.98 (108.48–186.74)			26.93 (18.20–40.71)		
	T0′	Mean ± SD	178.18 ± 66.08			244.14 ± 108.92			88.49 ± 32.50		
		Median (Q1–Q3)	153.64 (128.56–236.34)			243.63 (158.09–296.31)			83.07 (56.32–106.88)		
Recreational (*n* = 11)	T0	Mean ± SD	155.75 ± 54.74	−0.55	0.12	152.89 ± 56.79	−0.76	0.032	24.32 ± 8.03	−0.76	0.03
		Median (Q1–Q3)	129.51 (119.10–186.57)			158.62 (126.78–174.19)			24.78 (16.85–30.80)		
	T0′	Mean ± SD	196.80 ± 71.71			258.33 ± 117.61			94.66 ± 33.99		
		Median (Q1–Q3)	158.72 (144.69–251.18)			259.87 (160.95–289.86)			87.94 (67.78–120.16)		
Trained (*n* = 10)	T0	Mean ± SD	161.74 ± 72.17	0.02	1.00	165.83 ± 103.71	−0.64	0.073	40.92 ± 23.56	−0.64	0.08
		Median (Q1–Q3)	140.74 (132.07–165.05)			166.27 (87.17–204.23)			41.09 (20.86–59.00)		
	T0′	Mean ± SD	157.69 ± 55.67			228.52 ± 102.37			81.69 ± 31.06		
		Median (Q1–Q3)	152.30 (104.30–199.47)			220.85 (139.74–296.90)			81.53 (53.55–102.22)		

Values are presented as Mean ± Standard deviation (SD) and Median (Q1–Q3), where Q1 and Q3 represent the first and third quartiles. r = rank-biserial correlation coefficient, with around 0.1 considered weak, around 0.3 moderate, and ≥0.5 strong. *p* < 0.05 was considered statistically significant. T0—pre-supplementation period; T0′—post-supplementation period; AA—arachidonic acid; EPA—eicosapentaenoic acid; DHA—docosahexaenoic acid.

## Data Availability

The original contributions presented in the study are included in the article, further inquiries can be directed to the corresponding author.

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
