# Peer review of "Effects of 30-Day High-Dose Omega-3 Fatty Acid Supplementation on Plasma Oxidative Stress Enzyme Activities in Recreational and Trained Runners: A Pilot Study"

_nutrients, 2025, doi:10.3390/nu17182985_

Round 1
Reviewer 1 Report
Comments and Suggestions for Authors
General Comments
I found this manuscript to be an ambitious and timely pilot study exploring the effects of 30-day high-dose omega-3 supplementation on oxidative stress enzyme activity in recreational and trained runners. The study addresses an interesting and underexplored question, and the combination of biochemical markers with exercise-induced stress testing is commendable. The paper is generally well written, with a solid theoretical introduction and a clear structure.
However, several issues undermine the strength of the work. First, the rationale for using such a high dosage of omega-3 (9 g/day) requires more vigorous justification, especially given the potential risks of lipid peroxidation. Second, the sample size is small (n=21), which significantly limits statistical power and makes regression analyses unstable. Third, the interpretation of findings sometimes drifts toward overstatement—for instance, suggesting performance implications that were not measured. The methodology is detailed but certain aspects, such as dietary control, supplementation compliance, and the timing of blood sampling, deserve more transparency. Finally, the discussion would benefit from a deeper engagement with conflicting literature, especially regarding whether antioxidant supplementation may blunt training adaptations.
In sum, the manuscript has potential but requires revisions to temper its claims, expand on methodological transparency, and strengthen its discussion of limitations.
Specific Comments by Line Numbers
Comment 1 – Abstract (lines 14–31)
The abstract claims that supplementation “may enhance” antioxidant defense in trained individuals. While the cautious wording is appreciated, the results actually showed decreases in enzyme activity at baseline. This discrepancy should be clarified so the abstract reflects the nuanced findings.
Comment 2 – Introduction (lines 55–84)
The authors state that omega-3 fatty acids stimulate antioxidant systems, but the literature also highlights pro-oxidative risks of high PUFA intake. Although mentioned briefly, this tension deserves fuller treatment to justify the study’s hypothesis and dosage.
Comment 3 – Introduction (lines 109–115)
The authors claim no prior study has examined high-dose omega-3 supplementation in athletes. This may be true, but the introduction could benefit from acknowledging related studies in clinical populations or at lower doses to more precisely establish the novelty.
Comment 4 – Participants (lines 117–124)
Only healthy males were included. This limitation should be emphasized earlier, as it significantly reduces generalizability. Why were women excluded?
Comment 5 – Study design (lines 131–173)
The design is clear, but the decision to collect blood samples only at 24h and 48h post-exercise may have missed critical acute oxidative responses. The discussion (later) acknowledges this, but the methodological rationale should be given here.
Comment 6 – Supplementation details (lines 150–157)
Participants consumed 12 capsules in a single morning bolus. This approach could affect absorption and metabolic kinetics. A brief justification or reference for using bolus intake (rather than divided doses) would be helpful.
Comment 7 – Statistical analysis (lines 205–215)
Given the small sample, the reliance on multiple t-tests increases type I error risk. A repeated-measures model might have been more appropriate. At a minimum, a note on correction for multiple comparisons is needed.
Comment 8 – Results: EPA/DHA changes (lines 247–263)
The results show significant increases in DHA, particularly in recreational runners. The speculation about “utilization efficiency” is interesting but remains speculative without mechanistic data. This claim should be toned down or supported with more substantial evidence.
Comment 9 – Results: Enzyme activities (lines 279–305)
The reported decreases in GPx and SOD are central findings, but the framing as “stimulation” or “modulation” is misleading. The authors should describe this more directly as downregulation.
Comment 10 – Discussion (lines 330–332)
The conclusion that high-dose supplementation “leads to decreased antioxidant enzyme activities” is consistent with the data, but the next step—inferring this may benefit training adaptation—is speculative. A more cautious tone is warranted.
Comment 11 – Discussion (lines 366–383)
The suggestion that omega-3 supplementation may “compromise training effects” by lowering GPx is intriguing but highly speculative. No performance outcomes were measured. The discussion should clearly separate biochemical observations from hypothetical performance implications.
Comment 12 – Discussion (lines 392–411)
The interpretation that decreased SOD activity represents “reduced need for acute upregulation” is one possible explanation, but others (e.g., suppression of defense capacity) are equally plausible. This should be discussed as an open question rather than a firm conclusion.
Comment 13 – Discussion (lines 417–434)
CAT activity results are explained as timing-related, but this is a post-hoc rationalization. A more balanced acknowledgment would be that the study design may not have been sensitive enough to detect changes.
Comment 14 – Conclusions (lines 436–447)
The conclusions should highlight the pilot nature and emphasize the need for larger, controlled trials. Currently, the tone suggests broader generalizability than is justified.
Comment 15 – Limitations (lines 448–454)
The limitations section is brief. It should explicitly mention (1) the small sample size, (2) the absence of female participants, (3) the lack of dietary control, and (4) the risk of type I/II statistical errors.
Reviewer 2 Report
Comments and Suggestions for Authors
The manuscript explores the effects of 30-day high-dose omega-3 supplementation on antioxidant enzyme activities in recreational and trained runners. This is a novel and timely topic, as the interplay between polyunsaturated fatty acid supplementation, oxidative stress, and exercise adaptation remains under investigation. The paper is generally well written and the methodology is clearly described. However, there are several major concerns that limit the strength of the conclusions. Addressing these will considerably improve the manuscript.
1] The absence of a placebo-controlled group is a significant limitation. Without this, it is difficult to attribute the observed changes in antioxidant enzyme activities specifically to omega-3 supplementation. Please address this in both the limitations section and the discussion, and temper the conclusions accordingly.
2] The final sample (n=21) is very small, especially once divided into recreational and trained subgroups. This limits statistical power and increases the risk of type I/II errors. Please provide a justification for the chosen sample size, or acknowledge more strongly how this affects interpretation.
3] Blood sampling only at 24h and 48h post-exercise may have missed the acute oxidative stress response. Please justify the choice of these time points and discuss how this may influence your ability to capture relevant changes.
4] The exercise test (2800m + 400m sprint) is not standardized in literature; reproducibility across studies may be difficult.
Round 2
Reviewer 1 Report
Comments and Suggestions for Authors
Thank you for incorporation of my comments
Reviewer 2 Report
Comments and Suggestions for Authors
The authors have appropriately acknowledged the main limitations of their study, including the small sample size, male-only design, absence of randomization and placebo control, limited blood sampling time points, and the use of a non-standardized exercise protocol.